# Deep Network for the Integrated 3D Sensing of Multiple People in Natural Images

**Andrei Zanfir**[2] **Elisabeta Marinoiu**[2] **Mihai Zanfir**[2] **Alin-Ionut Popa**[2] **Cristian Sminchisescu**[1,2]
{andrei.zanfir, elisabeta.marinoiu, mihai.zanfir, alin.popa}@imar.ro,
cristian.sminchisescu@math.lth.se
[1]Department of Mathematics, Faculty of Engineering, Lund University
[2]Institute of Mathematics of the Romanian Academy

## Abstract

We present MubyNet – a feed-forward, multitask, bottom up system for the integrated localization, as well as 3d pose and shape estimation, of multiple people in monocular images. The challenge is the formal modeling of the problem that intrinsically requires discrete and continuous computation, e.g. grouping people vs. predicting 3d pose. The model identifies human body structures (joints and limbs) in images, groups them based on 2d and 3d information fused using learned scoring functions, and optimally aggregates such responses into partial or complete 3d human skeleton hypotheses under kinematic tree constraints, but without knowing in advance the number of people in the scene and their visibility relations. We design a multi-task deep neural network with differentiable stages where the person grouping problem is formulated as an integer program based on learned body part scores parameterized by both 2d and 3d information. This avoids suboptimality resulting from separate 2d and 3d reasoning, with grouping performed based on the combined representation. The final stage of 3d pose and shape prediction is based on a learned attention process where information from different human body parts is optimally integrated. State-of-the-art results are obtained in large scale datasets like Human3.6M and Panoptic, and qualitatively by reconstructing the 3d shape and pose of multiple people, under occlusion, in difficult monocular images.

## 1 Introduction

The recent years have witnessed a resurgence of human sensing methods for body keypoint estimation [1; 2; 3; 4; 5; 6; 7] as well as 3d pose and shape reconstruction [8; 9; 10; 11; 12; 13; 14; 15; 16; 17; 18; 19; 20; 21]. Some of the challenges are in the level of modeling – shifting towards accurate 3d pose and shape, not just 2d keypoints or skeletons –, and the integration of 2d and 3d reasoning with automatic person localization and grouping. The discrete aspect of grouping and the continuous nature of pose estimation makes the formal integration of such computations difficult. In this paper we propose a novel, feedforward deep network, supporting different supervision regimes, that predicts the 3d pose and shape of multiple people in monocular images. We formulate and integrate human localization and grouping into the network, as a binary linear integer program with optimal solution based on learned body part compatibility functions constructed using 2d and 3d information. State-of-the-art results on Human3.6M and Panoptic illustrate the feasibility of the proposed approach.

**Related Work.** Several authors focused on integrating different human sensing tasks into a single-shot end-to-end pipeline[22; 23; 24; 13]. The models are usually designed to handle a single person and often rely on a prior stage of detection. [13] encode the 3d information of a single person inside a feature map forcing the network to output 3d joint positions for each semantic body joint at its

corresponding 2d location. However, if some joints are occluded or hard to recover, their method may not provide accurate estimates. [9] use a discretized 3d space around the person, from which they read 3d joint activations. Their method is designed for one person and cannot easily be extended to handle large, crowded scenes. [25] use Region Proposal Networks [26] to obtain human bounding boxes and feeds them to predictive networks to obtain 2d and 3d pose. [27] provide a framework for the 3d human pose and shape estimation of multiple people. They start with a feedforward semantic segmentation of body parts, and 3d pose estimates based on DMHS [22], then refine the pose and shape parameters of a human body model [12] using non-linear optimization based on semantic fitting – a form of feedback. In contrast, we provide an integrated, yet feedforward, bottom up deep learning framework for multi-person localization as well as 3d pose and shape estimation.

One of the challenges in the visual sensing of multiple people is grouping – identifying the components belonging to each person, their level of visibility, and the number of people in the scene. Our network aggregates different body joint proposals represented using both 2d and 3d information, to hypothesize limbs, and later these are assembled into person instances based on the results of a joint optimization problem. To address the arguably simpler, yet challenging problem of locating multiple people in the image (but not in 3d) [1] assign their network the task of regressing an additional feature map, where a slice encodes either the $x$ or $y$ coordinate of the normalized direction of ground-truth limbs. The information is redundant, as it is placed on the 2d projection of the ground truth limbs, within a distance $\sigma$ from the line segment. Such part affinity fields are used to represent candidate part detection associations. The authors provide several solutions, one that is global but inefficient (running times of 6 minutes/image being common) and one greedy. In the greedy algorithm, larger human body hypotheses (skeletons) are grown incrementally by solving a series of intermediate bipartite matching problems along kinematic trees. While the algorithm is efficient, it cannot be immediately formalized as a cost function with global solution, it relies solely on 2d information, and the affinity functions are handcrafted. In contrast, we provide a method that leverages a volumetric 3d scene representation with learned scoring functions for the component parts, and an efficient global linear integer programming solution for person grouping with kinematic constraint generation, and amenable to efficient solvers (e.g. 30ms/image).

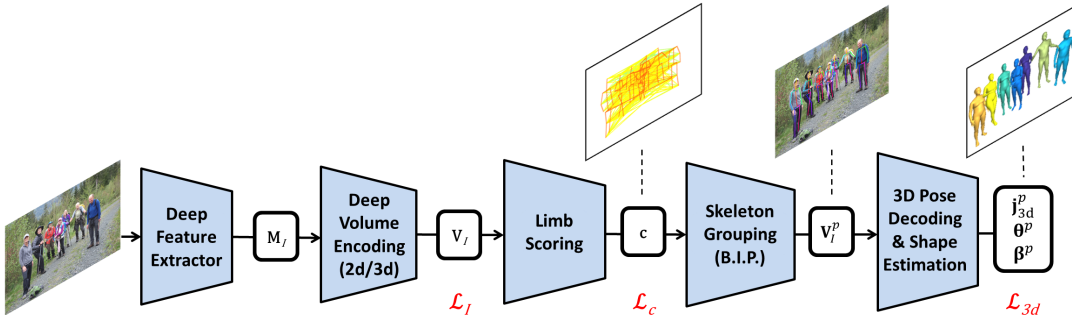

Figure 1: Our multiple person sensing pipeline, MubyNet. The model is feed-forward and supports simultaneous multiple person localization, as well as 3d pose and shape estimation. Multitask losses constrain the output of the *Deep Volume Encoding*, *Limb Scoring* and *3D Pose & Shape Estimation* modules. Given an image $I$, the processing stages are as follows: *Deep Feature Extractor* to compute features $\mathbf{M}_I$, *Deep Volume Encoding* to regress volumes containing 2d and 3d pose information $\mathbf{V}_I$. *Limb Scoring* collects all possible kinematic connections between 2d detected joints given their type, and predicts corresponding scores $\mathbf{c}$, *Skeleton Grouping* performs multiple person localization, by assembling limbs into skeletons, $\mathbf{V}_I^p$, by solving a binary integer linear program. For each person, the *3D Pose Decoding & Shape Estimation module* estimates the 3d pose and shape $(\mathbf{j}_{3d}^p, \boldsymbol{\theta}^p, \boldsymbol{\beta}^p)$.

## 2    Methodology

Our modeling and computational pipeline is shown in fig.1. The image is processed using a deep convolutional feature extractor to produce a representation $\mathbf{M}_I$. This is passed to a deep volume encoding module containing multiple 2d and 3d feature maps concatenated as $\mathbf{V}_I = \mathbf{M}_I \oplus \mathbf{M}_{2d} \oplus \mathbf{M}_{3d}$. The volume encoding is passed to a limb scoring module that identifies different human body

joint hypotheses and their type in images, connects ones that are compatible (can form parent-child relations in a human kinematic tree), and assigns them scores [1] given input features sampled in the deep volume encoding $\mathbf{V}_I$, along the spatial direction connecting their putative image locations. The resulting scores are assembled in a vector $\mathbf{c}$ and passed to a binary integer programming module that optimally computes skeletons for multiple people under kinematic constraints. The output is the original dense deep volume encoding $\mathbf{V}_I^p$, annotated now with additional person skeleton grouping information. This is passed to a final stage, producing 3d pose and shape estimates for each person based on attention maps and deep auto-encoders.

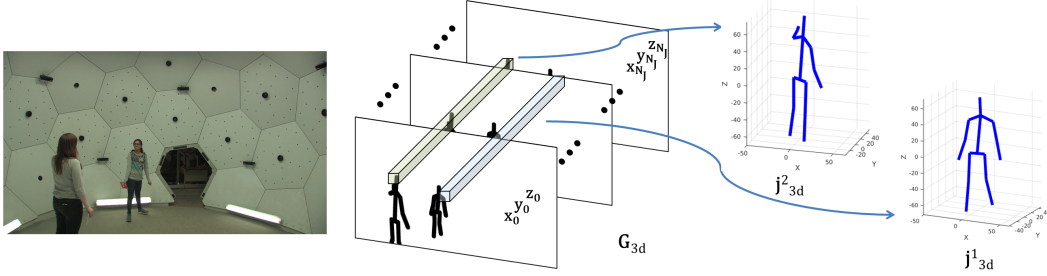

Figure 2: The volume encoding of multiple 3d ground truth skeletons in a scene. We associate a slice (column) in the volume to each one of the $N_J \times 3$ joint components. We encode each 3d skeleton $\mathbf{j}_{3d}^p$ associated to a person $p$, by writing each of its components in the corresponding slice, but only for columns 'intercepting' spatial locations associated with the image projection of the skeleton.

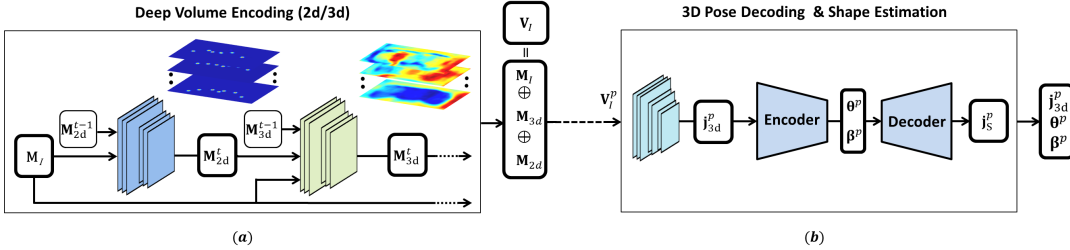

Figure 3: (*a*) Detailed view of a single stage $t$ of our multi-stage *Deep Volume Encoding (2d/3d)* module. The image features $\mathbf{M}_I$, as well as predictions from the previous stage, $\mathbf{M}_{3d}^{t-1}$ and $\mathbf{M}_{2d}^{t-1}$, are used to refine the current representations $\mathbf{M}_{3d}^t$ and $\mathbf{M}_{2d}^t$. The multi-stage module outputs $\mathbf{V}_I$, which represents the concatenation of $\mathbf{M}_I$, $\mathbf{M}_{2d} = \sum_t \mathbf{M}_{2d}^t$ and $\mathbf{M}_{3d} = \sum_t \mathbf{M}_{3d}^t$. (*b*) Detail of the *3D Pose Decoding & Shape Estimation* module. Given the estimated volume encoding $\mathbf{V}_I$, and the additional information from the estimation of person partitions $\mathbf{V}_I^p$, we decode the 3d pose $\mathbf{j}_{3d}^p$. By using auto-encoders, we recover the model pose and shape parameters $(\boldsymbol{\theta}^p, \boldsymbol{\beta}^p)$.

Given a monocular RGB image $I \in \mathbb{R}^{h \times w \times 3}$, our goal is to recover the set of persons $P$ present in the image, where $(\mathbf{j}_{2d}^p, \mathbf{j}_{3d}^p, \boldsymbol{\theta}^p, \boldsymbol{\beta}^p, \mathbf{t}^p) \in P$ with $1 \leq p \leq |P|$, $\mathbf{j}_{2d} \in \mathbb{R}^{2N_J \times 1}$ is the 2d skeleton, $\mathbf{j}_{3d} \in \mathbb{R}^{3N_J \times 1}$ is the 3d skeleton, $(\boldsymbol{\theta}, \boldsymbol{\beta}) \in \mathbb{R}^{82 \times 1}$ is the SMPL [12] shape embedding, and $\mathbf{t} \in \mathbb{R}^{3 \times 1}$ is the person's scene translation.

## 2.1 Deep Volume Encoding (2d/3d)

Given the resolution of the input image, the resolution of the final maps produced by the network is $H \times W$ with the network resolution of final maps $h \times w$. For the 2d and 3d skeletons, we adopt the Human3.6M [28] representation, with $N_J = 17$ joints and $N_L = 16$ limbs. We refer to $\mathbf{K} \in \{0, 1\}^{N_J \times N_J}$ as the kinematic tree, where $\mathbf{K}(i, j) = 1$ means that nodes $i$ and $j$ are endpoints of a limb, in which $i$ is the parent and $j$ the child node. We denote by $\mathbf{M}_I \in \mathbb{R}^{h \times w \times 128}$ the image

features, by $\mathbf{M}_{2d} \in \mathbb{R}^{h \times w \times N_J}$ the 2d human joint activation maps, and by $\mathbf{M}_{3d} \in \mathbb{R}^{h \times w \times 3N_J}$ a volume that densely encodes 3d information.

We start with a deep feature extractor (e.g. VGG-16, module $\mathbf{M}_I$). Our pipeline progressively encodes volumes of intermediate 2d and 3d signals (i.e. $\mathbf{M}_{2d}$ and $\mathbf{M}_{3d}$) which are decoded by specialized layers later on. An illustration is given in fig. 3. The processing in modules such as *Deep Volume Encoding* is multi-stage [29; 22]. The input and the results of previous stages of processing are iteratively fed into the next, for refinement. We use different internal representations than [29; 22], and rely on a single supervised loss at the end of the multi-stage chain of processing, where outputs (activation maps) are fused and compared against ground truth. We found this approach to converge faster and produce slightly better results than the more standard per-stage supervision [29; 22].

We construct a representation combining 2d and 3d information capable of encoding multiple people in the following way: given an input image $I$, our *Deep Volume Encoding* module produces an output tensor $\mathbf{M}_{3d}$ containing the 3d structure of all people present in the image. At training time, for any ground-truth person $p$ with $\mathbf{g}_{2d}^p$, $\mathbf{g}_{3d}^p$ joints, kinematic tree structure $\mathbf{K}$, and limbs $L_{2d} = \{(i,j)|\mathbf{K}(i,j) = 1, i,j \in \mathbb{N}, 1 \leq i,j \leq N_j\}$, we define a ground-truth volume $\mathbf{G}_{3d}$. For all points $(x,y)$ on the line segment of any limb $l \in L_{2d}$ connecting joints in $\mathbf{g}_{2d}^p$, we set $\mathbf{G}_{3d}(x,y) = (\mathbf{g}_{3d}^p)^\top$. The procedure is illustrated in fig. 2. This module also produces $\mathbf{M}_{2d}$, which encodes 2d joints activations. The corresponding ground-truth volume, $\mathbf{G}_{2d}$, will be composed of confidence maps, one for each joint type aggregating Gaussian peaks placed at the corresponding 2d joint positions. The loss function will then measure the error between the predicted $\mathbf{M}_{3d}, \mathbf{M}_{2d}$ and the ground-truth $\mathbf{G}_{3d}, \mathbf{G}_{2d}$

$$\mathcal{L}_I = \sum_{1 \leq x \leq h, 1 \leq y \leq w} \rho_{2d}(\mathbf{M}_{2d}(x,y), \mathbf{G}_{2d}(x,y)) + \sum_{\substack{1 \leq x \leq h, 1 \leq y \leq w \\ \mathbf{G}_{3d}(x,y) \text{ is valid}}} \rho_{3d}(\mathbf{M}_{3d}(x,y), \mathbf{G}_{3d}(x,y)) \quad (1)$$

We choose $\rho_{2d}$ to be the squared Euclidean loss, and $\rho_{3d}$ the mean per-joint 3d position error (MPJPE). We explicitly train the network to output redundant 3d information along the 2d projection of a person's limbs. In this way, occlusion, blurry or hard-to-infer cases do not negatively impact the final, estimated $\{\mathbf{j}_{3d}^p\}$, in a significant way.

## 2.2 Skeleton Grouping

Our skeleton grouping strategy relies on detecting potential human body joints and their type in images, assembling putative limbs, scoring them using trainable functions, and solving a global, optimal assignment problem (binary integer linear program) to find all connected components satisfying strong kinematic tree constraints – each component being a different person.

### Limb Scoring

By accessing the $\mathbf{M}_{2d}$ maps in $\mathbf{V}_I$ we extract, via non-max suppression, $N$ joint proposals $J = \{i|1 \leq i \leq N\}$, with $t$ an index set of $J$ such that if $i \in J$, $t(i) \in \{1,\ldots,N_J\}$ is the joint type of $i$ (e.g. shoulder, elbow, knee, etc.). The list of all the feasible kinematic connections (i.e. *limbs*) is then $L = \{(i,j)|\mathbf{K}(i,j) = 1, i,j \in \mathbb{N}, 1 \leq i,j \leq |J|\}$. One needs a function to assess the quality of different limb hypotheses. In order to learn the scoring $\mathbf{c}$, an additional network layer *Limb Scoring* is introduced. This takes as input the volume encoding map $\mathbf{V}_I$, and passes the result through a series of Conv/ReLU operations to build a map $\mathbf{M}_c \in \mathbb{R}^{h \times w \times 128}$. A subsequent process over $\mathbf{M}_c$ and $\mathbf{M}_{2d}$ builds the candidate limb list $L$ by sampling a fixed number $N_{samples}$ of features from $\mathbf{M}_c$ for every $l \in L$, along the 2d direction of $l$. The resulting features have dimensions $N_L \times N_{samples} \times 128$, and are passed to a multi-layer perceptron head followed by a softmax non-linearity to regress the final scoring $\mathbf{c} \in [0,1]^{N_L \times 1}$. Supervision is applicable on outputs of *Limb Scoring*, via a cross-entropy loss $\mathcal{L}_c$. Any dataset containing 2d annotations of human body limbs can be used for training.

### People Grouping as a Binary Integer Programming Problem

The problem of identifying the skeletons of multiple people is posed as estimating the optimal $L^* \subseteq L$ such that graph $G = (J, L^*)$ has properties *(i)* any connected component of $G$ falls on a single person, *(ii)* $\forall p, q \in L^*$, $p = (i_1, j_1)$ and $q = (i_2, j_2)$ with $t(i_1) = t(i_2)$ and $t(j_1) = t(j_2)$: if $j_1 = j_2$ then

$i_1 \neq i_2$ and if $i_1 = i_2$ then $j_1 \neq j_2$ – these constraints ensure that connected components select at most one joint of a given type, and *(iii)* the connected components are as large as possible.

Computing $L^*$ is equivalent to finding a binary indicator $\mathbf{x} \in \{0,1\}^{|L| \times 1}$ in the set $L$. We can encode the kinematic constraints *(ii)*, by iterating over all limbs $p \in L$, and finding all the limbs $q \in L$ that connect the same type of joints as $p$, and also share an end-point with it. Clearly, for any $p$, the solution $\mathbf{x}$ can select at most one of these limbs $q$. This can be written row-by-row, as a sparse matrix $\mathbf{A} \in \{0,1\}^{|L| \times |L|}$, that constrains $\mathbf{x}$ such that $\mathbf{A}\mathbf{x} \leq \mathbf{b}$, where $\mathbf{b}$ is the all-ones vector $\mathbf{1}_{|L|}$. In order to satisfy requirement *(i)*, we need a cost that well qualifies the semantic and geometrical relationships between elements of the scene, learned as explained in the **Limb Scoring** paragraph above. The limb score $\mathbf{c}(l), \forall l \in L$ measures how likely is that $l$ is a limb of a person with the particular joint endpoint types. To satisfy requirement *(iii)*, we need to encourage the selection of as many limbs as possible while still satisfying kinematic constraints. Given all these, the problem can naturally be modeled as a binary integer program

$$\mathbf{x}^*(\mathbf{c}) = \arg\max_{\mathbf{x}} \mathbf{c}^\top \mathbf{x}, \text{ subject to } \mathbf{A}\mathbf{x} \leq \mathbf{b}, \mathbf{x} \in \{0,1\}^{N_L \times 1} \tag{2}$$

where an approximation to the optimal $\mathbf{c}^* = \arg\max_{\mathbf{c}} \mathbf{x}^*(\mathbf{c})^\top \mathbf{g}_c$ is learned within the *Limb Scoring* module. At testing time, given the learned scoring $\mathbf{c}$, we apply (2) to compute the final, binarized solution $\mathbf{x}^*$ obeying all constraints. The global solution is very fast and accurate taking in the order of milliseconds per images with multiple people, as shown in the experimental section.

## 2.3 3d Pose Decoding and Shape Estimation

An immediate way of decoding the $\mathbf{M}_{3d}$ volume into 3d pose estimates for all the persons in the image, is to simply average 3d skeleton predictions at spatial locations given by the limbs selected by the people grouping module. However, this does not take into account the differences in joint visibility. We propose to learn a function that selectively attends to different regions in the $\mathbf{M}_{3d}$ volume, when decoding the 3d position for each joint. Given the feature volume $\mathbf{V}_I^p$ of a person and its identified skeleton, we collect a fixed number of samples along the direction of each 2d limb (see fig. 3 (b)). We train a multilayer perceptron to assign a score (weight) to each sample and for each 3d joint. The final predicted 3d skeleton $\mathbf{j}_{3d}^p$ is the weighted sum of the 3d samples encoded in $\mathbf{M}_{3d}^p$. The loss $\mathcal{L}_{3d}^p$ is computed as the MPJPE between $\mathbf{j}_{3d}^p$ and the ground truth skeleton $\mathbf{g}_{3d}^p$.

In order to further represent the 3d shape of each person, we use a SMPL-based human model representation [12] controlled by a set of parameters $\boldsymbol{\theta} \in \mathbb{R}^{72 \times 1}$, which encode joint angle rotations, and $\boldsymbol{\beta} \in \mathbb{R}^{10 \times 1}$ body dimensions, respectively. The model vertices are obtained as a function $\mathbf{V}(\boldsymbol{\theta}, \boldsymbol{\beta}) \in \mathbb{R}^{6890 \times 3}$, and the joints as the product $\mathbf{j}_s = \mathbf{R}\text{vec}(\mathbf{V}) \in \mathbb{R}^{3N_J \times 1}$, where $\mathbf{R}$ is a regression matrix and $\mathbf{V}$ is the matrix of all 3d vertices in the final mesh. Our goal is to map the predicted $\mathbf{j}_{3d}$ into a pair $(\boldsymbol{\theta}, \boldsymbol{\beta})$ that best explains the 3d skeleton. Previous approaches [14] formulated this task as a supervised problem of regressing $(\boldsymbol{\theta}, \boldsymbol{\beta})$, or forcing the projection of $\mathbf{j}_s$ to match 2d estimates $\mathbf{j}_{2d}$. The problem is at least two-fold: 1) $\boldsymbol{\theta}$ encode axis-angle transformations that are cyclic in the angle and not unique in the axis, 2) regression on $(\boldsymbol{\theta}, \boldsymbol{\beta})$ does not balance the importance of each parameter (e.g., the global rotation encoded in $\boldsymbol{\theta}_0$ is more important than the right foot rotation, encoded in a $\boldsymbol{\theta}_i$) in correctly inferring the full 3d body model. To address such difficulties, we model the problem as unsupervised auto-encoding inside a deep network, where the code is $(\boldsymbol{\theta}, \boldsymbol{\beta})$, and the decoder is $\mathbf{R}\text{vec}(\mathbf{V}(\boldsymbol{\theta}, \boldsymbol{\beta}))$, which is a specialized layer. This is a natural approach, as the loss is then simply $\mathcal{L}_{3d}^s = \rho_{3d}(\mathbf{j}_{3d}, \mathbf{j}_s)$, which does not force $\boldsymbol{\theta}$ to have a unique or a specific value, and naturally balances the importance of each parameter. Additionally, the task is unsupervised. The encoder is a simple MLP with ReLUs as non-linearities. To account for the unnatural twists along limb directions that may appear, and the fact that $\boldsymbol{\beta}$ is not unique for a 3d skeleton (it is unique only for a given $\mathbf{V}$), we also include in the loss function a GMM prior on the $\boldsymbol{\theta}$ parameters, and a L2 prior on $\boldsymbol{\beta}$. For those examples where we have 'ground-truth' $(\boldsymbol{\theta}, \boldsymbol{\beta})$ parameters available, they are fitted in a supervised manner using images and their corresponding shape and pose targets. The 3d loss can be derived as

$$\mathcal{L}_{3d} = \mathcal{L}_{3d}^s + \mathcal{L}_{3d}^p \tag{3}$$

## 3 Experiments

We provide quantitative results on two datasets, Human3.6m [28] and CMU Panoptic [30] as well as qualitative 3d reconstructions for complex images.

**Human3.6m** [28] is a large, single person dataset with accurate 2d and 3d ground truth obtained from a motion capture system. The dataset contains 15 actions performed by 11 actors in a laboratory environment. We provide results on the withheld, official test set which has over 900,000 images, as well as on the Human80K test set [31]. Human80K is a smaller, representative subset of Human3.6m. It contains 80,000 images (55, 144 for training and 24, 416 for testing). While this dataset does not allow us to assess the quality of sensing multiple people, it allows us to extensively evaluate the performance of our single-person component pipeline.

**CMU Panoptic** [30] is a dataset that contains multiple people performing different social activities (e.g. society games, playing instruments, dancing) in an indoor dome where multiple cameras are placed. We will only consider the monocular case for both training and testing. The setup is difficult due to multiple people interacting, partial views and challenging camera angles. Following [27], we test our method on 9, 600 sequences selected from four activities: *Haggling*, *Mafia*, *Ultimatum*, and *Pizza* and two cameras, 16 and 30 (we run the monocular system on each camera, independently, and the total errors are averaged).

**Training Procedure**. We use multiple datasets with different types of annotations for training our network. Specifically, we first train our $\mathbf{M}_{2d}$ component on COCO [32] which contains a large variety of images with multiple people in natural scenes. Then, we use the Human80K dataset to learn the 3d volume $\mathbf{M}_{3d}$, once $\mathbf{M}_{2d}$ is fixed. Human80K contains accurate 3d joint positions annotations, however limited to single person, fully visible images, recorded in a laboratory setup. We further use the CMU Panoptic dataset for fine-tuning the $\mathbf{M}_{3d}$ component on images with multiple people and for a large variety of (monocular) viewpoints, which results in occlusions and partial views. Based on the learned 2d joint map activations $\mathbf{M}_{2d}$, the 3d volume $\mathbf{M}_{3d}$ and the image features $\mathbf{M}_I$, we proceed to learn the limb scoring function, $\mathbf{c}$. For this task we use the COCO dataset. The attention-based decoding is learned on CMU Panoptic since having multiple people in the same scene helps the decoder understand the difficult poses and learn where to focus in the case of occlusions and partially visible people. Finally, Human80K is used to learn the 3d shape auto-encoder due to its variability in body pose configurations. We use a Nesterov solver with a learning rate of $1e - 5$ and a momentum of $0.9$. Our models are implemented in Caffe [33].

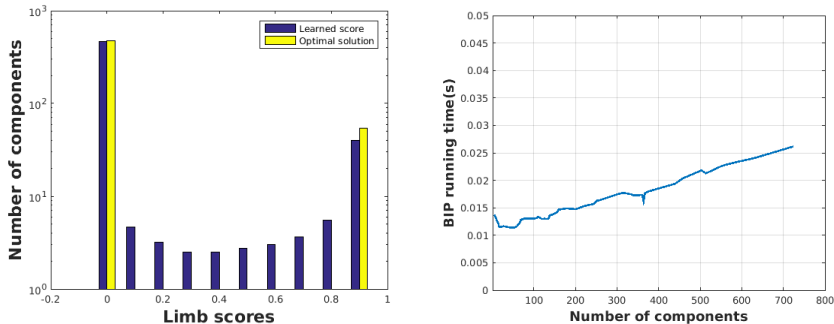

Figure 4: **(Left)** The distribution of the learned score $\mathbf{c}$ compared to the distribution of the selected limbs $\mathbf{x}^*$ after optimizing (2). Note that in the learned limb probability scores there are already many components close to $0$ or $1$ which indicates a well tuned function. **(Right)** Running time of our Binary Linear Integer Programming solver as a function of the number of components, $\dim(\mathbf{x})$. Notice fast running times for global solutions.

**Experimental Results**. Table 2 (left) provides results of our method on Human80K. Firstly, we show the performance without the attention-based decoding (simply averaging 3d skeletons at 2d locations in $\mathbf{M}_{3d}$ volume). This setup already performs better than DMHS [22]. Note that DMHS uses ground truth person bounding boxes while our method runs on full images. Our method with attention-based decoding obtains state-of-the art results on Human80K dataset. We also provide results on the withheld test set of Human3.6m (over 900,000 images), where we considerably improve over the state-of-the art (60 mm compared to 73 mm error). Results are detailed for each action in table 1. For the CMU Panoptic dataset, results are shown for each action in table 2 (right). When our method uses only Human80K as supervision for the 3d task, it already performs better than [27]. In

table 2 (right) we also show results with a fine-tuned version of our method on the Panoptic dataset. We sampled data from the *Haggling*, *Mafia*, *Ultimatum* actions (different recordings than those in test data) and from all cameras for fine-tuning our model. We obtained a total of $74,936$ data samples where the number of people per image ranges from $1$ to $8$. Our fine-tuned method improves the previous results by a large margin – notice errors of 72.1 mm down from 150.3 mm.

We show visual results of our method on natural images with complex interactions in fig. 5. We are able to correctly identify all the persons in an image as well as their associated 3d pose configuration, even when they are far in depth (first row) or severely occluded (last four rows).

**Limb Scoring**. In fig. 4 (left) we show the distribution of the learned scores $\mathbf{c}$ for the kinematically admissible putative limbs and the distribution of the optimal limb indicator vector components $\mathbf{x}^*$ over 100 images from the COCO validation set. The learned limb scoring already has many of its components close to either $0$ or $1$, although a considerable number still are 'undecided' resulting in non-trivial integer programming problems. We tested the average time taken by the binary integer programming solver as a function of the number of detected limbs (the length of the score vector $\mathbf{c}$). Figure 4 (right) shows that the method scales favorably with the number of components, i.e. $\dim(\mathbf{x})$.

| Method | A1 | A2 | A3 | A4 | A5 | A6 | A7 | A8 | A9 | A10 | A11 | A12 | A13 | A14 | A15 | Mean |
|--------|----|----|----|----|----|----|----|-----|-----|-----|-----|-----|-----|-----|-----|------|
| **[22]** | 60 | 56 | 68 | 64 | 78 | 67 | 68 | 106 | 119 | 77 | 85 | 64 | 57 | 78 | 62 | 73 |
| **[27]** | 54 | 54 | 63 | 59 | 72 | 61 | 68 | 101 | 109 | 74 | 81 | 62 | 55 | 75 | 60 | 69 |
| **MubyNet** | **49** | **47** | **51** | **52** | **60** | **56** | **56** | **82** | **94** | **64** | **69** | **61** | **48** | **66** | **49** | **60** |

Table 1: Mean per joint 3d position error (in mm) on the Human3.6M dataset. MubyNet improves the state-of-the-art by a large margin for all actions.

| Method | MPJPE(mm) |
|--------|-----------|
| **[22]** | 63.35 |
| **MubyNet** | 59.31 |
| **MubyNet attention** | **58.40** |

| Method | Haggling | Mafia | Ultimatum | Pizza | Mean |
|--------|----------|-------|-----------|-------|------|
| **[22]** | 217.9 | 187.3 | 193.6 | 221.3 | 203.4 |
| **[27]** | 140.0 | 165.9 | 150.7 | 156.0 | 153.4 |
| **MubyNet** | 141.4 | 152.3 | 145.0 | 162.5 | 150.3 |
| **MubyNet fine-tuned** | **72.4** | **78.8** | **66.8** | **94.3** | **72.1** |

Table 2: Mean per joint 3d position error (in mm). **(Left) Human80K dataset.** Our method with a mean decoding of the 3d volume obtains state-of-the art results. Adding the attention mechanism further improves the performance. **(Right) CMU Panoptic dataset.** Our method with 3d supervision only from Human80K performs better than previous works. Fine-tuning on the CMU Panoptic dataset drastically reduces the error.

# 4    Conclusions

We have presented a bottom up trainable model for the 2d and 3d human sensing of multiple people in monocular images. The proposed model, MubyNet, is multitask and feed-forward, differentiable, and thus conveniently supports training all component parameters. The difficult problem of localizing and grouping people is formulated as a binary linear integer program, and solved globally and optimally under kinematic problem domain constraints and based on learned scoring functions that combine 2d and 3d information for accurate reasoning. Both 3d human pose and shape are computed in a final predictive stage that fuses information based on learned attention maps and deep auto-encoders. Ablation studies and model component analysis illustrate the adequacy of various design choices including the efficiency of our global, binary integer linear programming solution, under kinematic constraints, for the human grouping problem. Our large-scale experimental evaluation in datasets like Human3.6M and Panoptic, and for withheld test sets of over 1 million samples, offers competitive results. Qualitative examples show that our model can reliably estimate the 3d properties of multiple people in natural scenes, with occlusion, partial views, and complex backgrounds.

**Acknowledgments:** This work was supported in part by the European Research Council Consolidator grant SEED, CNCS-UEFISCDI (PN-III-P4-ID-PCE-2016-0535, PN-III-P4-ID-PCCF-2016-0180), the EU Horizon 2020 grant DE-ENIGMA (688835), and SSF.

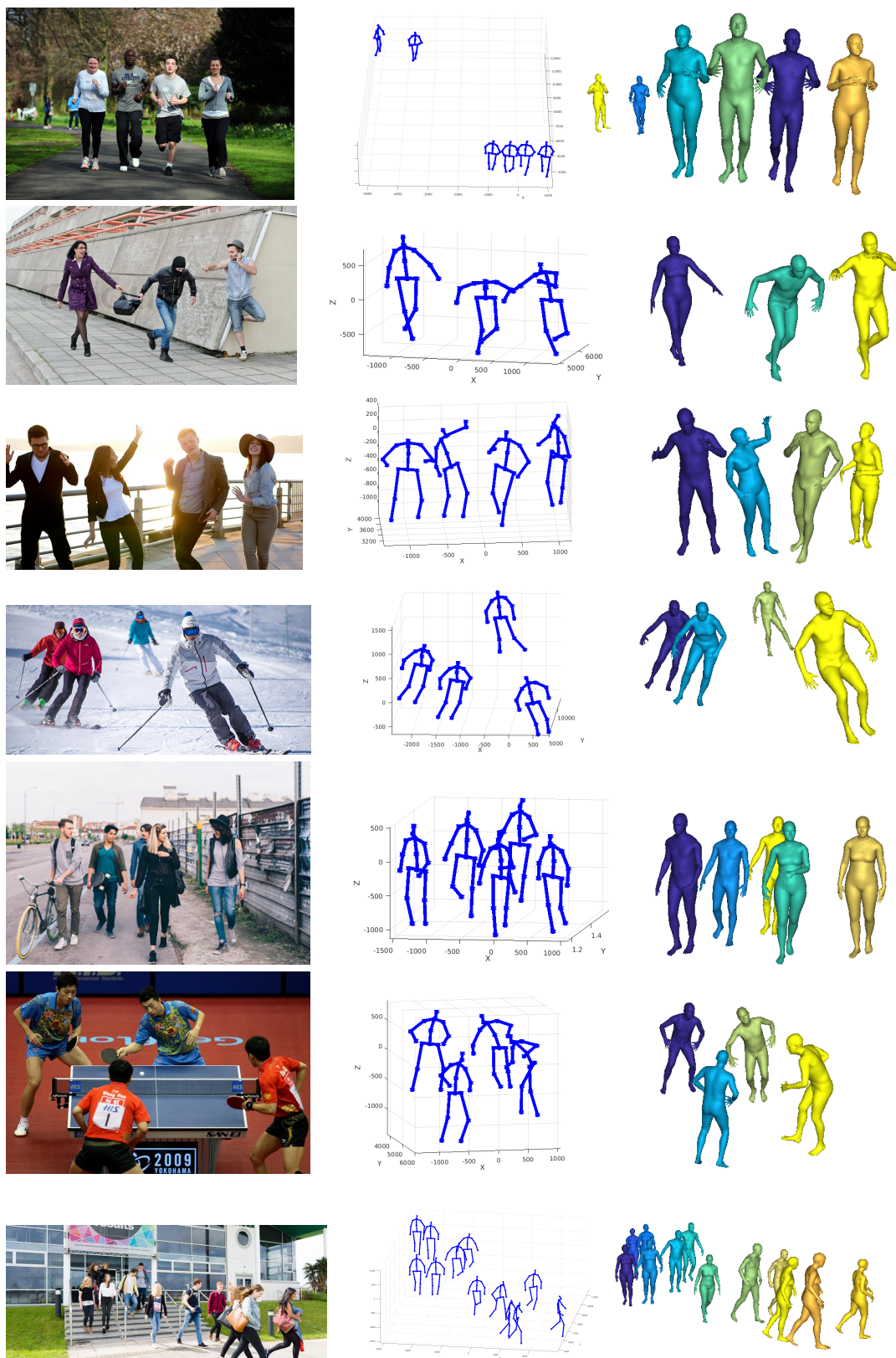

Figure 5: Human pose and shape reconstruction of multiple people produced by MubyNet illustrate good 3d estimates for distant people, complex poses or occlusion. For global translations, we optimize the Euclidean loss between the 2d joint detections and the projections predicted by our 3d models.

## Footnotes

[1]At this stage there is no person assignment so body joints very far apart have a putative connection as long as they are kinematically compatible.

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
