[Reviews · NeurIPS 2018]

Reviewer 1



This paper presents a system for 3D pose estimation of multiple people in monocular images. The system is end-to-end trainable, and was tested on two datasets, obtaining very impressive results. The pipeline has several steps, such as feature extraction and volume encoding, and uses techniques like deep learning and linear programming. === Strengths === 1) The main contribution of this paper is interesting and novel, and the experimental results provide strong evidence for the approach’s viability. 2) The paper is well-written. The contributions are clear and the main thesis of the paper is well-supported by the text and the experimental analysis. The approach is very well explained, with plenty of technical details. 3) The results are impressive. Two different datasets are used and the performance is compared against very recent results [21,22]. === Weaknesses === 1) The literature review (related work) discussion is extremely brief. Obviously, space on the paper is limited, but a bit more could be said about, for example, 3D pose estimation methods that use CNNs. Three citations missing are (and I’m sure there are more): * Li and Chan. 3D human pose estimation from monocular images with deep convolutional neural network. ACCV 2014. * Brau and Jiang. 3D human pose estimation via deep learning from 2D annotations. 3DV 2016. * Chen et al. Synthesizing training images for boosting human 3D pose estimation. 3DV 2016. 2) The novelty is mostly incremental. === Conclusion === This is a very good paper that presents a novel approach to a very important computer vision problem. The paper is well-written and the experimental results are impressive. Only concerns are limited literature review and incremental novelty.

Reviewer 2



This paper presents a multi-stage deep network for recovering 3D poses and shapes of multiple people from a single image. Unlike the top-down approaches that use human detectors to decompose the problem into single-person subtasks, this paper follows a recent bottom-up method that detects all the necessary local features so that the multi-person sensing problem can be solved jointly by graph matching. Given extracted features from the input image, the first stage predicts 2D keypoint heatmaps and 3D pose hypotheses on those limb pixels. The second stage first extracts local features from all the limb candidates and predicts their scores of being valid limbs or connections, and then solves a binary integer programming problem based on limb scores and the skeleton kinematic tree to group keypoints into 2D skeletons. In the third stage, the predicted 3D pose hypotheses on the predicted limb pixels are combined to synthesis the final 3D pose of each skeleton. The body shape is based on the SMPL model. The fitted shape and pose parameters (alpha and beta) are predicted from 3D skeletons via a differentiable mesh animation function. As the alpha and beta are unsupervised, GMM and L2 priors are used to regularize them. Results are first reported on a subset of Human3.6M with a single person in each image. Multi-person results are reported on CMU Panoptic datasets with comparisons to previous optimization-based methods. +This paper presents a complex deep network system for single-image 3D human sensing. All the components are clearly explained and corresponding chosen techniques are also well motivated, so the overall pipeline seems pretty reasonable. All the math notations are also well defined. +The limb scoring network is pretty interesting. Previous bottom-up approaches either predict dense limb vectors (PAFs) or some forms of per-pixel hough votes. The limb scoring network instead simplifies the problem into a light-weighted limb hypothesis verification problem based on sparsely sampled features. +The unsupervised SMPL modeling fitting is also interesting. +The good quantitative results on Human3.6M and CMU Panoptic datasets verify the effectiveness of the proposed method. -As it is a system-level work, such individual component may lack sufficient evaluations. For example, the limb scoring network bears some similarities with the part affinity fields in OpenPose. Although this paper argues that PAFs have some artificial parameters, e.g. the width of each PAF and need greedy grouping, it will be great to show some examples to compare the grouping quality and also present quantitative evaluations on 2D pose detection. I think it will strengthen the contribution and shed light on how better 2D pose detection can improve the overall results. The in-network SML fitting also needs some justifications. Experimental comparisons are need to demonstrate the claimed advantages over direct alpha-beta regression in [14]. -Some technical details may be missing. How is the person’s scene translation t being predicted? It is quite an important parameter. If the scene translations of two adjacent people are too close, there will be mesh penetration. But I didn’t see such results in the paper. I’m curious how this paper solved it or just didn’t present such results. Related to this question, since it is multi-person 3d pose detection, so how the 3D pose of each person is represented? The joint locations relative to root or joint locations in the scene? What is the value of N_samples? It is not clear from Fig. 3(b) that how the samples are extracted. A better figure is needed. -It will be great if this paper can present some failure cases which will inspire future works.

Reviewer 3



Summary: The paper presents a feed-forward trainable system for multiple human pose-related tasks of multiple people in monocular imagery. In particular, the approach considers the tasks of 2D localization, 3D pose and shape estimation. The approach consists of the following stages: image feature extraction, (redundant) 2D/3D pose encoding, limb scoring, people grouping, and 3D pose and shape estimation. The people grouping is realized as a linear binary integer program. State-of-the-art results are demonstrated on Human3.6M (official test set, contains a single person in each image) and Panoptic (contains multiple people). Pros: - novel with respect to the emerging literature on multi-person 3D sensing from monocular images - cyclic consistency ("auto-encoding") of 3D pose is an interesting idea - state-of-the-art results presented on suitable benchmarks; note, that more challenging ("in-the-wild" type benchmarks are sorely needed in the area) Cons: - the paper is frustratingly difficult to read in parts The major issue with the paper is related to the clarity of the presentation. In several key instances (e.g., Limb Scoring section), the manuscript passages are quite dense and terse rendering them difficult to follow. Also details are missing: - Line 31: "However, if some joints are occluded or hard to recover, their method may not succeed in providing accurate estimates." What exactly does this refer to? - Section 2.1: the description in the main body of the manuscript of the deep volume encoding via iterative processing is difficult to follow, more details are needed - How exactly is the solution to the binary integer program realized? - What is the source of the regressor, $\bf{R}$, in Section 2.3? - Section 2.3: How many samples are taken along the direction of each 2D limb? - Figure 5: What is the source of these images? Figure 2 and the corresponding caption are also difficult to follow. Maybe a simplified 2D figure would help. Unless the reader is familiar with extant state-of-the-art work on 2D and 3D pose estimation, parts of the paper are unnecessarily difficult to follow. It is suggested that Figure 5 be limited to three or four examples and the extra space used to address the clarity issues cited above. Empirical results: - What differences are there if any with the training data used for the proposed approach and the baselines? - What are the failure cases? Are the errors just due to inaccuracies in the estimates or are there cases of "catastrophic" failures, e.g., failure in predictions due to the 2D-3D ambiguities and people grouping (on Panoptic)? For completeness, why not also report the scene translation as in Table 1 [21] using the approach detailed in Fig. 5 caption? The authors have addressed my main concerns in their rebuttal regarding the evaluation and missing details. As a result, I have increased my rating to 8.